# Nearly Lattice-Matched GaN Distributed Bragg Reflectors with Enhanced Performance

**DOI:** 10.3390/ma15103536

**Published:** 2022-05-14

**Authors:** Ye Tian, Peng Feng, Chenqi Zhu, Xinchi Chen, Ce Xu, Volkan Esendag, Guillem Martinez de Arriba, Tao Wang

**Affiliations:** Department of Electronic and Electrical Engineering, The University of Sheffield, Sheffield S1 3JD, UK; ytian18@sheffield.ac.uk (Y.T.); pfeng3@sheffield.ac.uk (P.F.); czhu11@sheffield.ac.uk (C.Z.); xchen124@sheffield.ac.uk (X.C.); cxu31@sheffield.ac.uk (C.X.); vesendag1@sheffield.ac.uk (V.E.); gmartinezdearriba1@sheffield.ac.uk (G.M.d.A.)

**Keywords:** AlGaN/GaN, distributed Bragg reflector, electrochemical etching, nanoporous structure

## Abstract

Heavy silicon-doping in GaN generally causes a rough surface and saturated conductivity, while heavily silicon-doped *n++*-AlGaN with ≤5% aluminum can maintain an atomically flat surface and exhibit enhanced conductivity. Given this major advantage, we propose using multiple pairs of heavily silicon-doped *n++*-Al_0.01_Ga_0.99_N and undoped GaN instead of widely used multiple pairs of heavily silicon-doped *n++*-GaN and undoped GaN for the fabrication of a lattice-matched distributed Bragg reflector (DBR) by using an electrochemical (EC) etching technique, where the lattice mismatch between Al_0.01_Ga_0.99_N and GaN can be safely ignored. By means of using the EC etching technique, the *n++*-layers can be converted into nanoporous (NP) layers whilst the undoped GaN remains intact, leading to a significantly high contrast in refractive index between NP-layer and undoped GaN and thus forming a DBR. Our work demonstrates that the NP-Al_0.01_Ga_0.99_N/undoped GaN-based DBR exhibits a much smoother surface, enhanced reflectivity and a wider stopband than the NP-GaN/undoped GaN-based DBR. Furthermore, the NP-Al_0.01_Ga_0.99_N/undoped GaN-based DBR sample with a large size (up to 1 mm in width) can be obtained, while a standard NP-GaN/undoped GaN-based DBR sample obtained is typically on a scale of a few 100 μm in width. Finally, a series of DBR structures with high performance, ranging from blue to dark yellow, was demonstrated by using multiple pairs of *n++*-Al_0.01_Ga_0.99_N and undoped GaN.

## 1. Introduction

A distributed Bragg reflector (DBR), which consists of multiple pairs of two alternating layers with different refractive indices in each pair, is an indispensable component for a wide range of semiconductor photonic devices such as vertical-cavity surface-emitting lasers (VCSEL), where a microcavity can be formed by DBRs in a sandwiched configuration [1,2,3,4,5,6]. A DBR with high performance requires high reflectivity and a large stopband, which are both determined by contrast in refractive index between two alternating layers in each pair. It means higher contrast in the refractive index leads to higher reflectivity and a larger stopband. In order to achieve VCSEL with the best performance, it is ideal to have an epitaxially grown DBR, which in principle requires two alternating layers with lattice-matching and sufficiently high refractive index contrast in each pair. Thus far, great success has been achieved on epitaxially grown AlAs/GaAs based DBRs, as AlAs and GaAs exhibit large refractive index contrast (n = 2.89 and 3.42, respectively) and nearly lattice-matching between them, leading to GaAs-based VCSEL with high performance [1,2].

Unlike GaAs/AlAs based DBRs, it is a great challenge to achieve an epitaxially grown Al(Ga)N/GaN DBR with high performance for III-nitride based optoelectronics [7,8,9,10]. Firstly, the lattice mismatch between AlN and GaN (2.4%) is large enough to generate a high density of dislocations. Moreover, Al(Ga)N grown on GaN suffers from high tensile strain, potentially leading to a cracking issue. Secondly, the contrast in refractive index between GaN and AlN (~0.2) is far smaller than that for its AlAs/GaAs counterpart, meaning that a large number of pairs are required to obtain reasonably high reflectivity. This small contrast also means a narrow stopband. Finally, high-density dislocations generated as a result of the large lattice mismatch between AlN and GaN naturally extend to any overlying structures, thus potentially causing a degradation in the optical performance of an active region grown on top of the DBR.

In principle, AlInN alloy with proper indium or aluminium content can lattice-match GaN; for example, Al_0.82_In_0.18_N lattice matches GaN [11]. However, it is extremely difficult to obtain AlInN with high crystal quality whilst maintaining the required alloy content throughout the whole process of the DBR growth. InN has to be grown at a low temperature of <700 °C, while AlN is normally grown at a temperature of >1100 °C. Moreover, the large difference in interatomic spacing between AlN and InN (which is even larger than that between GaN and InN) leads to a severe miscibility gap, generating the well-known indium segregation and non-uniformity [12,13]. Another great concern is due to their even smaller refractive index contrast between AlInN and GaN than that between AlN and GaN [14].

One of the most promising approaches to achieving a lattice-matched DBR is to use multiple pairs of nanoporous (NP) GaN and GaN, where heavily silicon-doped GaN (i.e., *n++*-GaN) can be converted into NP-GaN by means of using electrochemical (EC) etching [15,16,17,18,19,20,21,22,23]. The mechanism for the EC etching is due to an initial oxidation process of GaN and a subsequent dissolution process in an acidic solution under bias. The injected holes initially oxidise GaN in an acidic electrolyte, and the oxidised layer is then chemically dissolved, leading to the formation of NP-GaN. Therefore, EC etching can be conducted only on highly conductive GaN (i.e., *n++*-GaN), while undoped GaN remains intact. Due to the formed porosities, the refractive index of NP-GaN is significantly lower than that of GaN [16,19,20,21,22,23]. For example, for a porosity of 0.5, the contrast in refractive index can be up to 0.6, which is even larger than that for AlAs/GaAs [23,24]. Therefore, by means of EC etching, the pairs of *n++*-GaN and undoped GaN can be converted into the pairs of NP-GaN and undoped GaN. Due to the large contrast in refractive index between NP-GaN and undoped GaN, a lattice-matched DBR is formed.

It is worth highlighting the EC etching technique sensitively depends on the conductivity of GaN. Moreover, it is necessary to have a very high doping level in order to fabricate a large area of DBR. However, generally speaking, *n++*-GaN tends to have saturated conductivity, limiting a fabricated DBR area to a small size, typically <200 µm in width [15]. Furthermore, *n++*-GaN leads to a rough surface, smearing the interface between *n++*-GaN and undoped GaN during their epitaxial growth. These fundamental issues form a major barrier to achieving a DBR with high performance in a large area.

It is understood that when the Al content in AlGaN is ≤5%, the heavily silicon-doped AlGaN not only maintains an atomically flat surface but also exhibits enhanced conductivity [25,26,27,28]. In this work, we propose flowing a tiny amount of Al precursor during the growth of *n++*-GaN, leading to the formation of heavily silicon-doped Al_0.01_Ga_0.99_N (i.e., *n++*-Al_0.01_Ga_0.99_N) instead of *n++*-GaN, where the lattice mismatch between Al_0.01_Ga_0.99_N and GaN, which is even smaller than that between AlAs and GaAs, can be safely ignored. As a result, the *n++*-Al_0.01_Ga_0.99_N exhibits an atomically smooth surface, leading to a sharp interface at NP-Al_0.01_Ga_0.99_N and undoped GaN after EC-etching and thus enhanced performance in terms of reflectivity and stopband compared with the NP-GaN/undoped GaN counterpart. Under identical bias used for EC-etching, the *n++*- Al_0.01_Ga_0.99_N shows enhanced conductivity by a factor of 2.5 compared with the *n++*-GaN. Furthermore, a large area DBR structure with ~ 1 mm in width was achieved.

## 2. Experimental Section

All the epi-wafers used in this work were grown on 2-inch (0001) sapphire substrates by means of the widely used two-step growth method using a metalorganic vapour phase epitaxy (MOVPE) technique. For all the epi-wafers, the growth of an initial GaN buffer was identical: after a sapphire substrate was subjected to an initial annealing process under H_2_ at a high temperature, a 25 nm low-temperature GaN nucleation layer was deposited, followed by the growth of a 2 µm GaN buffer layer at a high temperature. The subsequent growth was different for two kinds of samples labelled as Sample A and Sample B, respectively. For Sample A, the subsequent step was to grow 11 pairs of alternating *n++*-GaN layer and undoped GaN layer in each pair. For sample B, the subsequent step was carried out under almost identical conditions to those for sample A, and the only difference was to flow a small amount of Al precursor during the growth of the *n++*-GaN layer in each pair, while the growth of undoped GaN remains unchanged. As a result, 11 pairs of alternating *n++*-AlGaN layer (with a tiny amount of Al content) and undoped GaN layer were formed. Based on the Hall measurement, the doping concentrations of sample A and sample B are 4 × 10^19^ cm^−3^ and 4.75 × 10^19^ cm^−3^, respectively. Bruker D8 X-ray diffractometer was used to determine the Al content in *n++*-AlGaN, which is ~1%. This result is consistent with the data obtained from Energy-dispersive X-ray spectroscopy (EDX) measurements, where the EDX system was installed as part of a Raith 150 scanning electron microscopy (SEM) system. The SEM was also used to examine the surface morphology of the samples.

EC etching was performed at room temperature using a Metrohm Autolab PGSTAT204 system as a potentiostat to record the etching data. In order to carry out EC etching along a lateral direction (i.e., with injection current flowing along a lateral direction under bias), a SiO_2_ film with a thickness of 500 nm was deposited on each epi-wafer prior to EC etching by using a standard plasma-enhanced chemical vapor deposition (PECVD) technique. In this case, any EC etching along a vertical direction potentially via any defects such as either V-pits or screw dislocations could be eliminated. In order to enhance EC etching, parallel trenches with a spacing of 1 mm across a 2” epi-wafer were fabricated by means of a standard photolithography technique and then dry-etching processes using an inductively coupled plasma (ICP) technique. The trenches pass through all the 11 pairs of alternating *n++*-GaN (or *n++*-Al_0.01_Ga_0.99_N) layer and undoped GaN layer. In this case, all the *n++*-GaN (or *n++*-Al_0.01_Ga_0.99_N) layers were exposed to the air. Therefore, all the *n++*-GaN (or *n++*-Al_0.01_Ga_0.99_N) layers were exposed to an acidic electrolyte during subsequent EC etching processes. Finally, each epi-wafer was diced into a number of samples with a rectangular shape of 1 × 2 cm^2^. EC etching was conducted in 0.3 M nitric acid solution under 5.5 V bias, where indium contact was used as an anode and a platinum plate as a cathode. EC etching can only occur to highly conductive *n++*-GaN (or *n++*-Al_0.01_Ga_0.99_N) while undoped GaN remains intact. The EC etching processes were monitored by carefully observing the injection current. When the current approaches zero, it indicates that the etching processes are completed. Consequently, the pairs of *n++*-GaN (or *n++*-Al_0.01_Ga_0.99_N)/undoped GaN were converted into the pairs of NP-GaN (or NP-Al_0.01_Ga_0.99_N)/undoped GaN. It is worth highlighting that a higher current means the higher conductivity of a heavily silicon-doped layer under identical bias for EC-etching. After EC etching, the SiO_2_ on the top surface was simply removed with HF.

## 3. Methods

**Reflectivity measurements:** Reflectivity measurements were carried out by using a standard halogen lamp employed as a light source, which covers a wide spectral range from ultraviolet to far infrared. The reflected light was collected by an objective lens and was then introduced to a 50:50 beam splitter. Fifty per cent of the reflected light was collected by a CMOS camera to identify the position of the sample, while the other 50% of reflected light went to a Shamrock 500i Czerny–Turner monochromator through a fiber collimator and was finally detected by an air-cooled charge-coupled device (CCD). A commercial, calibrated mirror was used for calibration purposes.

**FDTD simulation:** Our simulation was carried out based on the structural data obtained from the cross-sectional SEM images, where the effective refractive indices for NP-Al_0.01_Ga_0.99_N and NP-GaN are 1.52 and 1.76, respectively, assuming the refractive index of GaN is 2.45. The electric field was injected by a plane wave source with an emission wavelength from 400 nm to 700 nm placed above the DBR. Reflectance was recorded by a frequency-domain power monitor. The simulation ran for 1000 fs with a minimum of 22 mesh points per wavelength, surrounded by perfectly matched layers (PML) used as boundary conditions.

## 4. Results and Discussion

Figure 1 shows the plan-view SEM images of Sample A and Sample B prior to EC etching, respectively. Figure 1a displays many V-pits on the surface of Sample A, which are generated due to heavy silicon-doping on GaN [24,25,26,27]. It also means that it is impossible to further increase the doping level by increasing a SiH_4_ flow rate (SiH_4_ is typically used as the precursor for silicon doping), as any further increase in SiH_4_ flow rate will deteriorate the surface completely. Certainly, it is expected that the appearance of a high density of V-pits will cause severe degradation in the performance of any further device structure grown on its top. In remarkable contrast, Figure 1b indicates that Sample B exhibits a very smooth, V-pit-free surface, although an identical SiH_4_ flow rate was used for both samples. The results further confirmed that an introduction of a small amount of Al precursor during the growth of highly silicon-doped GaN (leading to the formation of *n++*-Al_0.01_Ga_0.99_N in the present work) is beneficial for achieving a smooth, V-pit free surface. Of course, it is expected that the lattice mismatch between Al_0.01_Ga_0.99_N and GaN in each pair can be safely ignored.

Figure 2 shows the plan-view optical microscopy images of Sample A and Sample B after EC etching was conducted under identical conditions, respectively. In each case, due to the protection of the SiO_2_ layer on the sample surface, the EC etching starts from both sides of each trench and then spreads towards the centre. By carefully monitoring the injection current, Sample B shows the injection current as 2.5 times high as Sample A, which means that the conductivity of Sample B is enhanced by a factor of 2.5 compared with Sample A. Figure 2a also shows that Sample A exhibits an unetched part with a width of ~230 μm between two neighbouring trenches on average, while the unetched part of Sample B is much smaller compared with Sample A, only in a width of 38 µm on average. Once again, it has further confirmed that an introduction of 1% aluminium leads to significantly enhanced EC etching.

Figure 3 presents the cross-sectional SEM images of Sample A and Sample B after EC etching was conducted under identical conditions, respectively. In each case, all the 11 pairs of NP-layer and unetched GaN layer in both Sample A and Sample B were clearly observed. However, by carefully examining Figure 3, it can be found that the pores in Sample A are generally smaller than those in Sample B. Detailed image analysis (by using software), which was conducted on these cross-sectional SEM images, shows that the size of the pores in Sample A is 20 nm on average, leading to a porosity of 58%. Meanwhile, due to the existence of V-pits in Sample A, some parts of the heavily doped GaN layer in each pair could not be converted into pores by EC-etching. As a result, the interface between NP-GaN and undoped GaN in each pair is not so sharp. In contrast, by introducing 1% aluminium to the heavily doped GaN layers (i.e., Sample B), the pores in all the *n++*-Al_0.01_Ga_0.99_N layers become very uniform. Detailed image analysis indicates that the size of the pores in Sample B is 35 nm on average, giving a porosity of 71%. This also implies that the effective refractive index of the NP- Al_0.01_Ga_0.99_N in Sample B is smaller than that in Sample A, leading to enhanced contrast in the refractive index for Sample B compared with Sample A. Furthermore, the interface between the NP-Al_0.01_Ga_0.99_N layer and the undoped GaN layer in each pair for Sample B is much sharper than that in Sample A.

The effective index of the NP-(Al)GaN can be estimated from the volume average theory (VAT)
(1)$$n_{por}={\left[\;\left(1-\varphi \right)n_{\left(Al\right)GaN}^{2}+\varphi n_{air}^{2}\right]}^{1/2}$$
where $n_{por},\;n_{\left(Al\right)GaN},\;n_{air}$ and φ are the effective refractive index of the NP-(Al)GaN, the refractive index of intact GaN and the refractive index of air and porosity, respectively [28].

From Equation (1), the effective index of the NP-GaN for Sample A can be determined to be 1.76, while the effective refractive index of the NP-Al_0.01_Ga_0.99_N for Sample B is 1.52.

Figure 4 presents the reflectance spectra of Sample A and Sample B after EC etching; both were converted into DBR structures. For the measurement details, please refer to the Section 3. Figure 4a indicates 98.5% reflectivity at 529 nm as a central wavelength and a stopband of 146 nm for Sample A, while Sample B displays 99.9% reflectivity with a central wavelength at 525 nm and a stopband of 163 nm, as shown in Figure 4b. This has further confirmed that an introduction of a tiny amount of aluminium leads to a significant improvement in the optical performance of nanopores based on DBR. Figure 4 also includes their corresponding simulated reflectance spectra, which were obtained by using a standard finite-difference time-domain (FDTD) approach, where the effective refractive indices of the NP-GaN or the NP-Al_0.01_Ga_0.99_N used are 1.76 or 1.52 as given above, respectively. Figure 4 indicates that the simulated results agree with the measured reflectance spectra. For the simulation details, please refer to the Section 3. It is worth mentioning that the EC-etched NP-DBR shows a good uniformity over the sample surface when the etching conditions are fully optimised, which leads to a good uniformity in the reflectance. It will undoubtedly be beneficial to subsequent device applications.

Based on the result of Sample B, a number of NP-Al_0.01_Ga_0.99_N based DBR structures with different central wavelengths were obtained by tuning the individual thicknesses of *n++*-Al_0.01_Ga_0.9_N and undoped GaN in each pair and maintaining identical EC etching conditions. Figure 5a presents the reflectance spectra of a series of DBR mirrors with different central wavelengths at 485 nm, 508 nm, 550 nm and 565 nm, respectively. They all show high reflectivity (>99%) and a wide stopband (>110 nm). Accordingly, Figure 5b shows their optical images of these DBR structures. As the optical images were taken using a white lighting source, the colours from these DBR structures also represent the central wavelengths of these DBR structures, which are from blue through green/yellow to dark yellow. These also match the reflectance spectra of these DBR structures.

## 5. Conclusions

In summary, it is well-known that heavy silicon-doped GaN generally exhibits a rough surface and saturated conductivity, while heavily silicon-doped AlGaN with ≤5% aluminium can maintain an atomically flat surface and show enhanced conductivity. Given the major advantage of AlGaN with ≤5% aluminium, we proposed growing multiple pairs of *n++*-Al_0.01_Ga_0.99_N and undoped GaN instead of *n++*-GaN and undoped GaN, which can be converted into multiple pairs of NP-Al_0.01_Ga_0.99_N and undoped GaN after EC etching. Compared with an NP-GaN/undoped GaN-based DBR, such multiple pairs of NP-Al_0.01_Ga_0.99_N and undoped GaN exhibit enhanced high contrast in refractive index between NP-Al_0.01_Ga_0.99_N and undoped GaN, leading to a DBR structure with enhanced reflectivity and a wider stopband, where the lattice mismatch between Al_0.01_Ga_0.99_N and GaN can be safely ignored. More importantly, a much larger area of a DBR structure can be obtained after EC etching on the multiple pairs of *n++*-Al_0.01_Ga_0.99_N and undoped GaN than on the multiple pairs of *n++*-GaN and undoped GaN. Finally, this work demonstrated a series of DBR structures with high performance, ranging from blue to dark yellow, all based on *n++*-Al_0.01_Ga_0.99_N and undoped GaN.

## Figures and Tables

**Figure 1 materials-15-03536-f001:**
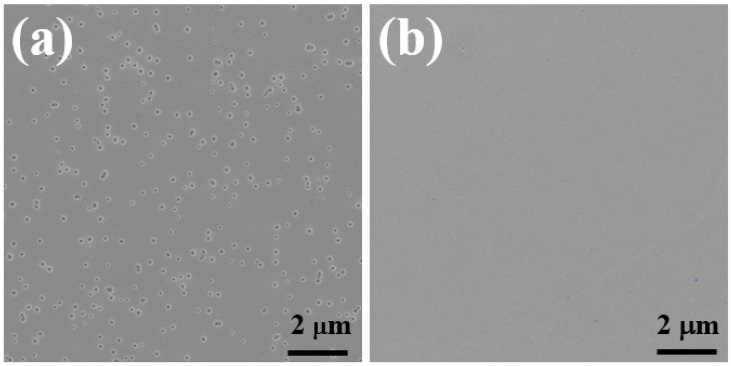
Plan-view SEM images of (**a**) sample A and (**b**) sample B prior to EC etching.

**Figure 2 materials-15-03536-f002:**
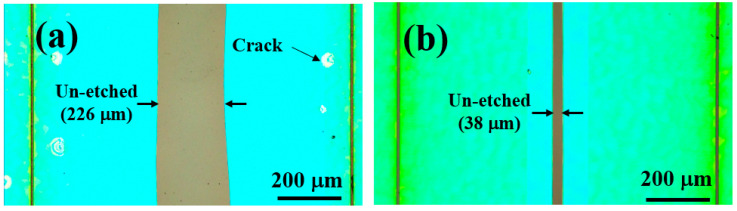
Plan-view optical microscopy images of (**a**) Sample A and (**b**) Sample B after EC etching (scale bar = 200 mm). The spacing between two neighbouring trenches is 1 mm.

**Figure 3 materials-15-03536-f003:**
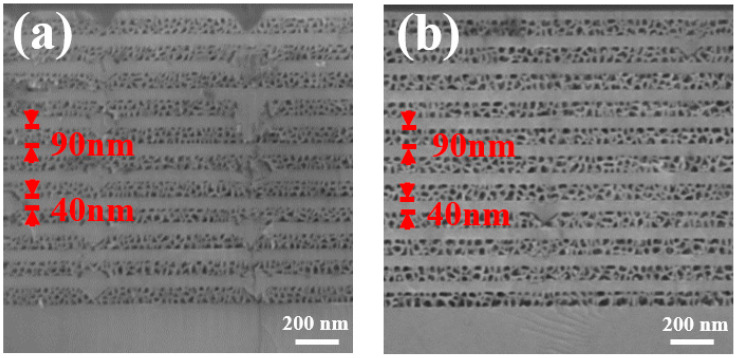
Cross-sectional SEM images of (**a**) Sample A and (**b**) Sample B after EC etching.

**Figure 4 materials-15-03536-f004:**
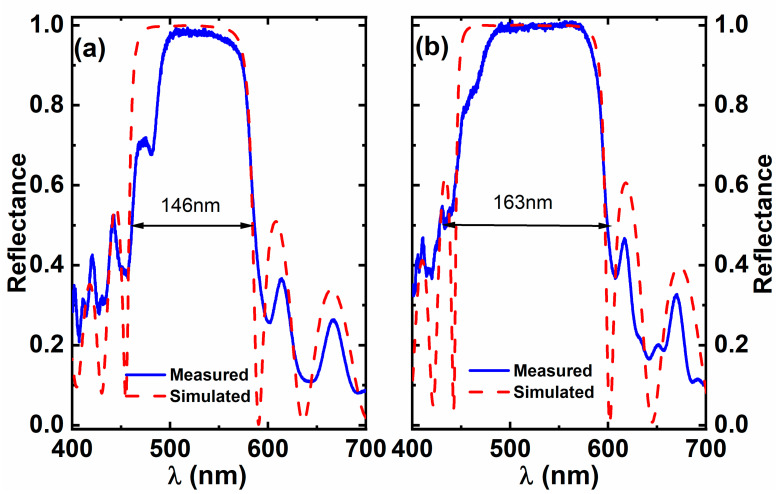
Reflectance spectra of (**a**) Sample A and (**b**) Sample B after EC etching, which were converted to the DBR structures.

**Figure 5 materials-15-03536-f005:**
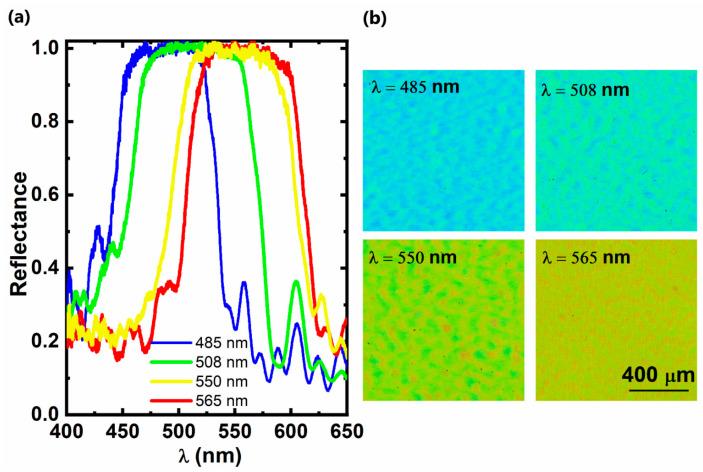
(**a**) Reflectance spectra from four different kinds of NP-Al_0.01_Ga_0.99_N/GaN DBR structures, each with a central wavelength at 485 nm, 508 nm, 550 nm and 565 nm, respectively; (**b**) Optical images of these NP-Al_0.01_Ga_0.99_N/GaN DBR structures taken using a white light source (scale bar = 400 μm).

## Data Availability

Data available on request.

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
