# Peer review of "Nearly Lattice-Matched GaN Distributed Bragg Reflectors with Enhanced Performance"

_materials, 2022, doi:10.3390/ma15103536_

Round 1

Reviewer 1 Report

The author has proposed a smooth surface, enhanced reflectivity, and a wider stopband by designing NP-Al0.01Ga0.99N/undoped GaN based DBR system. Challenging experimental results have been obtained through the research and their findings are in good agreement with the simulation. However, the language of the manuscript needs to be edited for engaged reading.

1) The author stated that the scale bar is 400 mm in Figure caption 5; however, it is 400 micrometer in the figure itself. The author should correct this.

2) The manuscript has some English problems. It has to be edited for fluent reading by avoiding repetitions of the similar sentence structures.

Author Response

1) The author stated that the scale bar is 400 mm in Figure caption 5; however, it is 400 micrometer in the figure itself. The author should correct this.

Answer:  Done.

2) The manuscript has some English problems. It has to be edited for fluent reading by avoiding repetitions of the similar sentence structures.

Answer: Thank you for the kind advice. It has been checked.

Reviewer 2 Report

The main problem, which is solved and clearly defined in introduction, is to improve the quality of distributed Bragg reflectors for nitride optoelectonic applications. AlGaN/GaN DBR are difficult to grow, and request a compromise between sufficient difference in refraction index and acceptable strain energy in AlGaN layers. To grow high quality interfaces is also difficult. Usage of porous GaN layers in Bragg reflectors was first time reported in 2015. For such DBR high n-type doping is necessary, which causes deterioration of interface morphology. This manuscript offers new, original way to solve this problem:  A high Si doping level can be achieved without deterioration of interface morphology and keeping low strain in DBR when very low Al content (1%) is used in doped layers.

The article is well written and easy to read. It could be presented in the way it is.

However, in case authors have available data it would be good to add the information about:

  • N-type doping level of GaN and AlGaN layers - It is important topic of the manuscript, the change in porous morphology can suggest change in doping level. The information, that doping level is above 1019cm-3is not sufficient. In case the authors know the doping level, please, specify them in the manuscript.
  • Inhomogeneity of reflectance spectra over the sample surface - I can imagine, that this should not harm in majority of applications. However, in scientific article I would suggest to evaluate this inhomogeneity and comment it with respect to different application of DBR.

Author Response

However, in case authors have available data it would be good to add the information about:

  • N-type doping level of GaN and AlGaN layers - It is important topic of the manuscript, the change in porous morphology can suggest change in doping level. The information, that doping level is above 1019cm-3 is not sufficient. In case the authors know the doping level, please, specify them in the manuscript.

Answer: The doping levels have been specified in the manuscript on page 3.

  • Inhomogeneity of reflectance spectra over the sample surface - I can imagine, that this should not harm in majority of applications. However, in scientific article I would suggest to evaluate this inhomogeneity and comment it with respect to different application of DBR.

Answer: The EC etched nanoporous DBR shows a good uniformity over the sample surface when the etching conditions are fully optimised. There is no obvious inhomogeneity in terms of reflectance spectra. This is undoubtedly beneficial to the subsequent devices employing the NP-DBRs. 

A comment is added on page 6.

Author Response

  1. Section 2. Result and discussion starts immediately after section 1. Introduction,
    which is not only unusual but in this case it can be easily fixed because the first two
    paragraphs of Section 2, describe the experimental procedures. If I may, I would like
    to suggest to change section 2 as: 2. Experimental Here include the top two paragraphs of the current section 2.
  2. The contents of 4. Methods may be added before the Result and discussion as: 3.
    Methods
  3. Then introduce, 4. Result and discussion from the 3rd Paragraph …Figure 1 shows
    the …… and Conclusion as the last section 5. Conclusion

Answer: The sections has been reorganised according to the suggestions.

  1. Figure 5 is in two parts: (a) and (b) in figure captions but (a) and (b) are not indicated
    in the figure. This should be changed and clarified.

Answer: Done.